# An eCoach-Pain for Patients with Chronic Musculoskeletal Pain in Interdisciplinary Primary Care: A Feasibility Study

**DOI:** 10.3390/ijerph182111661

**Published:** 2021-11-06

**Authors:** Cynthia Lamper, Ivan Huijnen, Maria de Mooij, Albère Köke, Jeanine Verbunt, Mariëlle Kroese

**Affiliations:** 1Department of Rehabilitation Medicine, Functioning, Participation & Rehabilitation, Faculty of Health, Medicine and Life Sciences, Care and Public Health Research Institute (CAPHRI), Maastricht University, 6200 MD Maastricht, The Netherlands; ivan.huijnen@maastrichtuniversity.nl (I.H.); m.demooij@maastrichtuniversity.nl (M.d.M.); albere.koke@maastrichtuniversity.nl (A.K.); jeanine.verbunt@maastrichtuniversity.nl (J.V.); 2Centre of Expertise in Rehabilitation and Audiology, Adelante, 6432 CC Hoensbroek, The Netherlands; 3Department of Health Services Research, Faculty of Health, Medicine and Life Sciences, Care and Public Health Research Institute (CAPHRI), Maastricht University, 6200 MD Maastricht, The Netherlands; marielle.kroese@maastrichtuniversity.nl

**Keywords:** chronic musculoskeletal pain, primary care, eHealth, blended care, interdisciplinary care, feasibility, mixed-methods design

## Abstract

eHealth could support cost-effective interdisciplinary primary care for patients with chronic musculoskeletal pain. This study aims to explore the feasibility of the eCoach-Pain, comprising a tool measuring pain complexity, diaries, pain education sessions, monitoring options, and chat function. Feasibility was evaluated (June–December 2020) by assessing learnability, usability, desirability, adherence to the application, and experiences from patients and general practitioners, practice nurses mental health, and physiotherapists. Six primary healthcare professionals (PHCPs) from two settings participated in the study and recruited 29 patients (72% female, median age 50.0 years (IQR = 24.0)). PHCPs participated in a focus group. Patient data was collected by evaluation questionnaires, individual interviews, and eCoach-Pain-use registration. Patients used the eCoach during the entire treatment phase (on average 107.0 days (IQR = 46.0); 23 patients completed the pain complexity tool and used the educational sessions, and 12 patients the chat function. Patients were satisfied with the eCoach-Pain (median grade 7.0 (IQR = 2.8) on a 0–10 scale) and made some recommendations for better fit with patient-specific complaints. According to PHCPs, the eCoach-Pain is of added value to their treatment, and patients also see treatment benefits. However, the implementation strategy is important for successful use of the eCoach-Pain. It is recommended to improve this strategy and involve a case-manager per patient.

## 1. Introduction

Chronic musculoskeletal pain (CMP) is a significant public health problem occurring in 19–28% of the European population [1,2]. It is expected that this number will increase in the next years, in line with an aging population [3]. The current health system for patients with CMP is fragmented, leading to high societal and healthcare costs [4,5,6]. Therefore, the World Health Organization (WHO) calls for a change in health systems focusing on interdisciplinary rehabilitation care and the improvement of self-management skills of patients on long term [7]. However, in order to reach this, there is a need for changes in knowledge, skills, and attitudes of healthcare professionals, as well as changes in the organization of healthcare.

Challenges in this change are accessibility and cost-effectiveness of rehabilitation care, for which eHealth can be a solution [8]. eHealth is defined as the use of information and communication technology for health [9,10]. A wide range of eHealth tools (such as mobile applications and online interventions) have been developed to improve self-management for acute and chronic pain, with promising results regarding their effectiveness [11,12,13]. Several reasons for the additional value of eHealth in the treatment for patients with CMP can be mentioned.

First, current care for chronic pain is fragmented and continuity of care for the individual patient is often lacking. eHealth can improve healthcare organization as it can facilitate communication and collaboration between healthcare professionals of different disciplines [14]. Accordingly, the WHO advises integrating rehabilitation care within and between primary (general practice), secondary (general hospital), and tertiary care (specialized care centers) [7]. They advise to implement eHealth to facilitate continuity of care in integrated health systems by stimulating daily activities and participation of patients, which are rehabilitation goals [15].

Second, currently, healthcare professionals receive training on diagnosis and treatment, primarily focused on knowledge within their own discipline [16]. This ranges from biomedical oriented care focusing on attempts to solve the pain, toward biopsychosocial oriented care which focuses on optimizing functioning despite pain [17,18]. However, the recommended approach by the WHO requires an integral biopsychosocial vision applied by all healthcare professionals. Currently, patients receive various treatment approaches causing confusion, resulting in unsuccessful organization of integrated care. An eHealth application can facilitate an integral vision on pain and a common language, which are components of integrated care [19]. In this way, it supports the treatment program of all participating healthcare professionals.

Third, earlier studies indicated that eHealth improves self-care support and improves daily activities for people with chronic illnesses [20,21,22]. eHealth, consisting of a combination of tools, might be of added value and useful as part of a blended care intervention. This is studied previously with separate tools for online pain education or keeping track of daily activities and participation in combination with face-to-face consultations [23,24]. The combination of tools is not studied previously and might lead to better informed and more actively involved patients with increased autonomy, as well as a shift of the role of the healthcare professional into adviser or coach [25]. Moreover, it is assumed that this blended care can stimulate integrated care in the long term and decrease healthcare costs [26,27].

To study the additional value of eHealth in an interdisciplinary network of healthcare professionals for patients with CMP, we implemented an electronic Coach (eCoach-Pain) to facilitate pain rehabilitation within the South East of the Netherlands [28,29]. Based on feedback in this earlier performed implementation, the eCoach-Pain is further improved into its current version. The eCoach-Pain aims to support the provision of integrated rehabilitation care with a shared biopsychosocial vision on health within the Network Pain Rehabilitation Limburg. Within this network, patients and Primary Health Care Professionals (PHCPs), existing of general practitioners (GPs), physiotherapists (PTs), and practice nurses mental health (PNMHs) use the eCoach-Pain. It comprises a measurement tool for assessing complexity of the pain problem, diaries, pain education sessions, monitoring options, and a chat function. Whether it is feasible to use in clinical practice is currently unknown. Therefore, this study aims to explore the feasibility of the eCoach-Pain for patients and PHCPs.

## 2. Materials and Methods

This study (June 2020 and December 2020) had a mixed-methods design. Feasibility was evaluated with a focus on learnability, usability, desirability, adherence to the application, and experiences from patients and PHCPS. These were measured by use of patient questionnaires, data about eCoach-Pain-use, a focus group with PHCPs, and interviews with patients. Ethical approval was obtained from the Medical Ethics Committee Z, the Netherlands (METCZ20190037). Patients did not have to pay for participation in Network Pain Rehabilitation Limburg or the eCoach-Pain. During a patient’s first login in the eCoach-Pain, an electronic informed consent for the use of the eCoach-Pain and consent for transferability of their contact details to the researcher were registered. Additionally, for the telephonic interview, patients were asked for informed consent and for recording the interview. PHCPs were asked for informed consent at the start of the focus group.

### 2.1. Sample and Setting

PHCPs (GPs, PNMHs, and PTs) of two interdisciplinary primary care practices were recruited to participate in this feasibility study (*n* = 6). They all participated in the Network Pain Rehabilitation Limburg (situated in the South East region of the province Limburg, The Netherlands). This network within and between primary, secondary, and tertiary care aims to support a shared biopsychosocial vision regarding CMP, early recognition of patients with subacute complaints, and a person-centered referral and treatment.

For the current project, patients were recruited by the participating PHCPs. They were eligible if they were ≥18 years at the start of the study, had CMP or musculoskeletal pain at increased risk of becoming chronic (based among criteria on the STarT MSK tool [30,31]), were willing to improve their functioning despite the pain, and had adequate Dutch literacy to use the eCoach-Pain. Exclusion criteria were pregnancy or any medical (orthopedic, rheumatic, or neurological) or psychiatric disease which could be treated by a more appropriate therapy, according to the expert opinion of the GP.

Once a new patient with CMP, or with an increased risk of developing chronic pain, consulted a PHCP, the patient was asked to use the eCoach-Pain. The PHCP gave the main instructions and sent a manual by email.

### 2.2. The eCoach-Pain

The eCoach-Pain has been designed by Sananet Care B.V., based on earlier developed eCoaches, such as for Inflammatory Bowel Disease and heart failure [29,32,33]. It contains different goals or opportunities for both patients with CMP and their PHCPs. For patients, the goal is to improve and maintain self-management in coping with pain. For PHCPs the goal is to facilitate biopsychosocial assessment for treatment planning and to monitor the treatment progress of patients with CMP. The eCoach-Pain has been developed in an iterative co-creative development process with the collaboration of researchers, technical experts, patients, and PHCPs. The results of this study will be published elsewhere [28]. The eCoach-Pain can be used on mobile phones, tablets, laptops, and PCs with internet connections.

#### 2.2.1. Application for the Patient

Each patient has an own account for the eCoach-Pain (Figure 1), which could be created in two different ways: First, the PHCP could create a patient’s account by filling in the patient’s contact details after which the patient receives two emails, one with an account name and one with a password. Consecutively, the patients’ account is automatically linked to that of his/her treating PHCP. Second, patients could do a self-subscription throughout a webpage. In this way, a patient is invited to complete contact details, to create a password, and to connect him/herself to his/her own PHCP. Subsequently, the patient’s username is sent to the patient by email.

After login by the patient, a home screen is presented, which contains four different elements (Figure 1).

1.The pain complexity tool:

The pain complexity tool supports the PHCPs in their decision-making for problem-mapping and treatment selection. It consists of two parts:

(A)The STarT MSK Tool assessing the complexity of the pain problem for referral within primary care. The patient’s first action in the eCoach-Pain is completing this questionnaire. The Dutch version of the STarT MSKTool is translated and validated [30,31]. The STarT MSK Tool exists of nine Yes (=1) or No (=0) questions regarding activity level, anxiety, depression, and thoughts about CMP and one Visual Analogue Scale (0–10) to assess pain intensity (0–4 = 0 points, 5–6 = 1 point, 7–8 = 2 points, 9–10 = 3 points). All scores are summed, and a total score of 0–4 indicated a low risk, a total score from 5–8 indicated a moderate risk, and a total score from 9–12 indicated a high risk of developing CMP.(B)To further differentiate within the range of primary, secondary and tertiary rehabilitation care an additional set of questions about the biopsychosocial complaints and background of the patient was added to be filled in by the PHCP. After completion of both parts of the complexity tool, the eCoach-Pain calculates the score and assigns the best-fitting referral option to assist the PHCP. The PHCP discusses the results with the patient and refers him/her to the most appropriate treatment via shared decision making.

2.Diaries:

The eCoach-Pain also contains the possibility to use diaries. The PHCP decides, together with the patient, if and how often diaries will be sent to the patient. Diaries can automatically be sent every week, every two weeks, or once a month. However, the diary option can also be neglected. The automatic setting of these diaries is once a week. The diaries exist of the pain complexity tool with additional questions. This extension exists of additional questions based on the questions in the STarT MSK tool scored with “Yes”. The answers to these additional questions could be discussed with the PHCP during consultation and used to adjust the treatment or to provide additional educational material to the patient.

3.Education sessions:

The educational sessions provide patients background information about topics related to pain and pain-related disability, such as the difference between acute and chronic pain, treatment of pain, biopsychosocial influences on their pain, information about work and pain, and treatment options. The educational sessions are interactive (YouTube videos and quiz questions with feedback on answers), and they are integrated to stimulate learning and improving knowledge about (chronic) pain. The educational materials are presented in 13 themes and per theme subdivided over several sessions (Figure 1).

4.Chat function:

The chat function is used to send bidirectional messages containing questions or treatment material between patient and PHCP. All communication between patient and PHCP remains accessible in the eCoach-Pain to enable patients to reread answers, advice, or treatment exercise at later moments and times.

#### 2.2.2. Application for the PHCPs

PHCPs could access the eCoach-Pain via a secured webpage on their own device. The PHCPs were instructed to monitor and analyze the patient’s situation within a few working days after the patient had completed the pain complexity tool or diary, and to respond as quickly as possible to messages from the patients. To facilitate interpretation of the pain complexity tool and diaries and to save PHCPs’ time, information within the application was supportively presented using overviews, graphs, and colored risk flags. Based on the results of the pain complexity tool, different flags appeared on the intervention list: a red flag for a high risk, an orange flag for a medium risk, and a green flag for a low risk for developing CMP (Figure 1).

PHCPs had only access to data of patients treated by themselves. It was possible that more PHCPs, for example, a PT and GP, have access to the data of the same patient in case it was a joint patient. When the PHCP sent a message to a patient or another PHCP, respectively, the other PHCPs and patient were able to read this message in the chat function.

An instruction meeting of one hour to become familiar with the possibilities of the eCoach-Pain was provided to all PHCPs before the start of the study. The software developers and research team facilitated this meeting. Afterward, a paper-copy instruction manual was provided. Moreover, during the pilot, the PHCPS could contact the service desk of the software developers when help was needed or technical issues occurred.

### 2.3. Data Collection and Analysis

#### 2.3.1. Learnability, Usability, and Desirability

In September 2020, when the PHCPs had used the eCoach-Pain already for approximately three months, the researcher sent a questionnaire to participating patients. The questionnaire assessed learnability (5-items), usability (5-items), and desirability (6-items) and was an adjusted version of a questionnaire used in a study by Hochstenbach et al. (2016) [34]. Usability was defined as ‘the extent to which the application could be used by patients with CMP to monitor their pain, physical activity, and participation level effectively, efficiently, and satisfactorily in everyday practice’. Learnability was defined as ‘the time and effort required for these patients to use the application’. Desirability was defined as ‘the extent to which the application was fun and engaging to use for these patients. Patients rated each item on a 1–5 Likert scale (completely disagree–completely agree); higher scores indicated better learnability, usability, and desirability. A separate item about the recommendation of the eCoach-Pain to family and friends on a 5-point scale, and a separate item about the overall acceptance of the eCoach-Pain for treatment purposes on a 10-point scale, were added.

Before data analysis, negatively-keyed items were reversed-scored using Microsoft Excel, version Professional Plus 2016, the Microsoft Corporation, Santa Rosa, CA, USA. Median scores with interquartile ranges per item and category were calculated. To identify differences between the PHCPs disciplines, a sub-analysis with discipline as dependent variable was performed.

#### 2.3.2. Adherence to the Application

To assess the patients’ adherence, process data from the pain complexity tool, diaries (filled out or not, time of fill out, answers), and from the educational sessions (opened or not, time of opening, how often opened) were logged on the server. The data collected between June 2020 and September 2020 were exported in October 2020.

Median scores and interquartile ranges were calculated using Microsoft Excel, for the number of days patients were active in the eCoach-Pain, number of completed pain complexity tools, diaries, educational sessions, and chat messages used.

Moreover, data about the PHCPs was collected. The median and interquartile ranges of the number of log-ins of the PHCPs was registered overall and per PHCP discipline.

#### 2.3.3. Experiences of Patients

Based on stratified probability sampling on sex, age, and PHCP, patients were contacted for a telephonic interview by the researcher to gain more insight into the experiences with the eCoach-Pain in September 2020. It was intended to ask approximately 16 patients, of which eight agreed, until data-saturation would be reached. However, as data saturation was not reached after this number of interviews, five additional interviews were performed in December 2020. Topics discussed in the semi-structured interviews included: the use and acceptance of the pain complexity tool, diaries, educational sessions, and chat function, the supportiveness of the application regarding self-management, and technological functioning of the application. Interviews were audio-recorded.

The audio recordings of the interviews were transcribed verbatim. These written interviews were independently analyzed with inductive and deductive thematic analysis by two researchers (C.L. and M.d.M.) using QSR International Pty Ltd. (2018) NVivo (Version 12), https://www.qsrinternational.com/nvivo-qualitative-data-analysis-software/home (accessed on 4 November 2021) [35,36,37]. Data were sorted based on pre-defined themes of the semi-structured interview-guide. Within these themes, sub-categories were created based on the data. After the first two interviews, main themes and codes were discussed and finalized. Thereafter, all other interviews were analyzed and discussed by adding additional codes under the predefined themes.

#### 2.3.4. Experiences of PHCPs

In September 2020, the researcher (C.L.) and observer (A.K.) held an online focus group interview with all participating PHCPs via Zoom [38]. Technical experts of Sananet B.V. (manufacturers of the eCoach-Pain) were available to take notes for future improvements. Before the start of the focus group, participating PHCPs completed questions about the topics on the agenda for the focus group. This encouraged them to formulate an individual opinion before the focus group started and to share this during the meeting. Topics discussed included: use and acceptance of the application, supportiveness of the application in monitoring, advising and treating patients, fit with daily care, and technical functioning of the application. The focus group was audio-recorded.

During the focus group with the PHCPs, the observer (A.K.) made notes and gave a summary per topic discussed. These summaries were asked to be confirmed by the PHCPs during the focus group. Before analysis, the audio recording was used to add additional notes to the summaries by the researchers (C.L. and M.d.M.). These summaries were independently analyzed with thematic analysis on the topics discussed by two researchers (C.L. and M.d.M.) using QSR International Pty Ltd. (Melbourne, Australia) (2018) NVivo (Version 12), https://www.qsrinternational.com/nvivo-qualitative-data-analysis-software/home (accessed on 4 November 2021) [35].

## 3. Results

Data was collected from 29 patients in total; see Table 1. They were eight male and 21 female participations aged between 24–71 years (Median = 50, IQR = 24). The GPs were the primary contact person for 14 patients, the PTs for 13 patients, and the PNMH for two patients. Sixteen patients used the self-registration webpage, while 13 patients were registered by their PHCP. The GP and PNMH of primary care practice one recruited, together, eight patients. The PT of primary care practice one did not recruit any patients. In primary care practice two, 21 patients were recruited by the GP, PNMH, and PT.

### 3.1. Learnability, Usability, and Desirability

Twenty-three patients received the invitation for the evaluation questionnaire (September 2020), of whom 11 patients responded (48%). The responders were older (65 (IQR = 23) years old) than the non-responders (48.0 (IQR = 24) years old), and there were less females than males (female responders: 55% (6 out of 11); female non-responders: 83% (10 out of 12)) compared to the total sample.

Six patients started using the eCoach-Pain after the questionnaire was sent and were therefore not invited. The patients who filled in the evaluation questionnaire were an average of 65 (IQR = 23) years old, 55% (6 out of 11) were female, and, on average, active in the eCoach-Pain for 109 (IQR = 41) days. Ten patients answered all questions, and one patient answered only the questions regarding learnability and usability.

Table 2 presents the overall median score (GP and PT), as well as the median score of the categories, and items separately per discipline (by a GP or PT). The scores show that patients learned quickly how to manage the application (Median = 5.0, IQR = 1.0) and could easily use the different components of the eCoach-Pain (Median = 5.0, IQR = 1.5). The desirability was scored with a median score of 4.0, IQR = 2.0. The overall acceptance, rated by the question “I would like to recommend the application to other patients” was scored with a median of 4.0 (IQR = 2.0). Patients gave the eCoach-Pain a total overall score of 7.0 (IQR = 2.8) on a 0–10 Numeric Rating Scale. Patients subscribed by GPs (*n* = 6) scored 5.0 (IQR = 0.0) for learnability, 5.0 (IQR = 1.0) for usability, and 4.5 (IQR = 2.0) for desirability, while patients subscribed by PTs (*n* = 5) scored 5.0 (IQR = 1.0), 5.0 (IQR = 2.0), and 4.0 (IQR = 1.0), respectively.

### 3.2. Adherence to the Application

At the end of October 2020, for 26 of the 29 patients (median age 53.5 years (IQR = 24.75), 69% female (18 out of 26)), the export data about adherence to the application was available. Three patients were asked to participate in the interviews in December 2020. At that moment, exports were already performed; therefore, no export data of them were available. At the moment of data export, the included 26 patients were, on average, 107.0 (IQR = 46.0) days active in the eCoach-Pain. Ten of them stopped using the eCoach-Pain prematurely because they finished treatment (*n* = 3) or did not want to use it anymore (*n* = 7). The other 16 patients were still active in the eCoach-Pain at that time.

Twenty-three patients completed the pain complexity tool (Median = 7.0, IQR = 3.0, 1x low risk, 16x medium risk, 7x high risk). For 21 patients, their PHCPs also answered the second part of the pain complexity tool. On average, the diaries were 6.0 (IQR = 3.5) times filled (*n* = 23).

The educational sessions were opened by 23 patients, and they read, on average, 12.0 (IQR = 5.0) educational sessions per person. On average, each separate educational session in the eCoach-Pain was started by 19.5 (IQR = 6.3) individual patients. In total, 224 unique educational sessions were opened by these patients. As there were 13 sessions, this means that some patients read (a part of) the educational sessions several times. Fourteen patients completed all education sessions.

Twelve messages were sent from seven (27%) unique patients to their PHCP, and five messages were sent from the PHCPs to the patients by the chat function. The patients started all conversations. They often elaborated on their diary answers, technical dysfunction of the eCoach-Pain, or they explained why they were not able to fill in the diaries.

The six PHCPs together logged in on average 6 times (IQR = 16.75), the GPs on average 16 times (IQR = 8), the PTs on average 35.5 times (IQR = 32.5), and the PNMHs on average 2.5 times (IQR = 1.5).

### 3.3. Experiences of Patients and PHCPs

At the end of September 2020, 16 patients were asked to participate in a telephonic interview, to which eight agreed. To reach data saturation, five additional patients had to be asked, of which three agreed to be interviewed in December 2020. This led to 11 patients participating in the telephonic interviews (mean duration 15 min). The participants had a median age of 60.0 (IQR = 2) years, 73% was female (8 out of 11), and they were active in the eCoach-Pain for 100.5 (IQR = 31.75) days.

Two GPs (one male and one female), two PTs (one male and one female), and one PNMH (one female) participated in a focus group. In addition, one other PNMH (one female) participated in an individual telephonic interview, as she was not able to participate in the focus group.

#### 3.3.1. Overall Opinion and Usage

Patients stated that they were positive about the eCoach-Pain because the functionality worked well for their treatment, it was easy to use, and text was written in clear and understandable language. The content was perceived as informative concerning their pain complaints and knowledge about pain pathophysiology. The interaction between patient and PHCP in the diaries and the quiz questions in the educational sessions of the eCoach-Pain were experienced of added value. Some patients appreciated the reminders for diaries and educational sessions as it gave them structure and control. However, for other patients, these automatic reminders were perceived as somewhat stressful.

Before patients (*n* = 6) started using the eCoach-Pain, they expected the content would be more tailored to their own medical complaints and history. Furthermore, patients expected that the eCoach-Pain would motivate them for treatment compliance to improve their complaints. Although PHCPs are able to change diary frequency, six patients expected less frequent diaries and repetition of information. Besides, some patients indicated that they had preferred to receive more information (by their PHCP or a pamphlet) about the content, frequency of questions, and expected duration of the eCoach-Pain program when they started to use it. Some patients expected more feedback from the eCoach-Pain itself about their answers or an automatic end-session in the eCoach-Pain to close it. Seven patients found it frustrating that, in their opinion, “non-relevant” questions kept returning. The option to indicate a holiday leave and stop sending reminders during this leave was felt to be missing.


*R16: “Basically, I think it is a good app. However, the questions appear too frequent, too standard.”*


Among the PHCPs, the eCoach-Pain was most often used by the PTs. One PT used it to structure the content of the treatment sessions and to deliver additional information to the patient.


*PT2: “I like the idea that every week new educational sessions about pain are open for the patient. And, that I can see what the patient answered, which information they have read, and that I can use that during the treatment session. This causes more structure in my treatments.”*


Another PT used it for educational purposes for the patient, as well, but did not use the results to guide or adjust treatment as the other PT did. In this feasibility study, the PNMHs hardly used the eCoach-Pain because it was not clear for them how to integrate it in their treatment. PNMHs perceived the eCoach-Pain options offered as specifically PTs treatment options. PTs registered most patients by themselves, which gave them control over the number of patients in the eCoach-Pain. Furthermore, controlling this registration facilitated the ability to inform patients before the start. In addition, the GPs used the eCoach-Pain to score the pain complexity assessment and to support the referral of the patient to the PT. They did not use it to offer treatment purposes or pain education to the patient. The patients that entered the study by a GP most often used the self-registration webpage. GPs indicated that this route was timesaving for them. GPs mentioned that the eCoach-Pain provided them an extra treatment option above the current treatment when they referred a patient to the PT.


*GP2: “The eCoach-Pain is an extra treatment option above the existing options. As a GP, it is important to know the content of the treatment options when referring to a PT, and it is great that we can offer something extra.”*


#### 3.3.2. Pain Complexity Tool and Diaries

Patients indicated difficulties in distinguishing the pain complexity tool from the diaries, as the tool and the diaries both were presented as a questionnaire in the eCoach-Pain. Therefore, in this paragraph, the tool and diaries are presented together. Eight of the 11 patients perceived the usability of the pain complexity tool and diaries as good. They indicated that the pain complexity tool and diaries were easy to use, not too time-consuming, easy to understand, and that the reminders by email were of added value. In addition, patients perceived the content as easy to keep track of their pain complaints and the amount of questions as good. However, most patients (*n* = 8) indicated that the repetition and frequency of the questions were too high. They also missed background information of the questions in the introduction of the eCoach-Pain.


*R02: “I thought that I had to fill in some questions a few times. However, the questions came every day or week for two or three months. And this was not explained to me beforehand.”*


Overall, most patients indicated that they perceived the questions in the pain complexity tool and diaries as less applicable in their situation. As several patients had co-morbidities besides CMP, it was difficult for them to know how to interpret the questions. For some questions, it was unclear for them whether the answers should be given with the perspective of having CMP, or from the perspective as a person having pain and other co-morbidities. For example, it was not always possible to indicate exactly their own pain complaints or to adjust answers to questions properly when their situation changed. Sometimes, the eCoach-Pain gave more insight into patients’ complexity and impact of their own complaints, which was perceived as heavy to encounter for some patients. Patients without difficulties in daily social participation or psychosomatic problems perceived answer options as less applicable in their specific situation. However, they understood that general questions were formulated for all different kinds of CMP.

The pain complexity tool was the most important tool for GPs in the eCoach-Pain. GP1 indicated that he used it to objectify referral and to get more insight into the complexity of the pain problem. However, GP1 indicated that the digital version directed the referral more than the paper-version. The eCoach-Pain automatically calculates the score and assigns the best-fitting referral option, while, with the paper-version, this can easily be overruled when necessary, according to the opinion of the GP.


*GP1: “When using the paper-version, you have more freedom in the choice of the treatment. As you can overrule the score of the patient easier. In the eCoach-Pain, the treatment options are more limited based on the answers of the patients. Which is a strength of the eCoach-Pain.”*


The two PTs used the pain complexity tools in combination with the diaries. For GP1, the graphical display of the results was especially of added value as it gave insight into the effect of the treatment. As improvement, all PHCPs indicated that the graphical displays of the diaries could be upgraded, as it was not always immediately clear for them if the patient’s score was positive or negative.

#### 3.3.3. Educational Sessions

The educational sessions were perceived as interesting with clarifying quiz questions and links to YouTube videos. The sessions about ‘What is pain’ and ‘Pain and being active’ were perceived as the most useful sessions. The sessions about work and work disability were not appropriate for every patient as some were retired or did not have a job. Two patients indicated that they desired more subjects and educational sessions, for example, about general health.


*R10: “It was a revelation for me, because through the information in the educational sessions, in addition with information on the same topic given by my PHCP, I understand how my brain controls the pain”.*


The usability and comprehensibility of the educational sessions were perceived as good as the language used was easy to understand. However, three patients found the language level even too easy and the repetition of subjects in the text as too much. One patient indicated that it was more useful for her when the sessions were not divided over several days, but that all sessions can be followed at once.

Overall, five of the eleven patients indicated that they did not receive new information in the educational sessions in comparison with what they already knew about pain (out of earlier treatments). Some other patients indicated that they perceived recognition and acceptance of their CMP during the sessions due to explanations about the pathophysiology of pain. One patient indicated the sessions as confronting as she/he recognized her/himself for the first time as a patient with chronic pain.


*R13: “I have read all sessions and the total overview was good for me. But at the same time it was also confronting, maybe that was good, as well.”*


The educational sessions were most often used by the PTs, and sometimes by the PNMH. PT1 used it to guide the content of his treatment, and PT2 and both PNMHs used it as additional education material for the patient. They indicated that patients were satisfied with the content of the educational sessions and that it gave them more insight into their pain problem. However, they perceived the educational sessions as less applicable for patients with a lower IQ-score or restricted literacy.

#### 3.3.4. Chat Function and Communication with PHCP

Two patients used the chat function, while nine patients indicated they did not. Those two patients were positive about its usability.

Four patients indicated that they had contact with their PT about the diaries and educational sessions they performed in the eCoach-Pain and rated these of added value. For at least one patient, the physiotherapy treatment was adjusted based on the results in the eCoach-Pain. Moreover, some patients discussed the diary questions about their psychosocial status. Furthermore, patients indicated that the pain education received by their PT fitted well with the information in the educational sessions. The eCoach-Pain resulted in a better patient-PT treatment relationship. Three patients had questions about the eCoach-Pain and needed extra support from their PHCP, for example, about the content of the eCoach-Pain, when to finish using the eCoach-Pain, or extra practical tips regarding their pain complaints. Moreover, some patients mentioned that they had discussed technical issues with their PHCP, such as logging in, bugs in the sessions, or difficulties with data exchange between the PHCP and patient.

The other seven patients mentioned no contact with their PHCP about their activities in the eCoach-Pain. Reasons for this ranged from patients’ holidays and sick leave periods, technical issues which limited eCoach-Pain-use, or the fact that the patient had not filled in the pain complexity tools and diaries before the next contact with the PHCP could take place.

Patients did not bother with the fact that their PHCP was able to track their activity in the eCoach-Pain, while some of them did not know this option before the interview. Two patients mentioned that they felt no need to discuss their activity in the eCoach-Pain with their PHCP. Most patients indicated that the possibility to discuss their activity online with the eCoach-Pain was of added value, especially in the situation of COVID-19 they were in during the pilot period, as live contact with PHCPs was only limited to emergency consultations.

PHCPs indicated that they did not use the chat function of the eCoach-Pain often as they preferred other ways to communicate with the patient, such as email, chat functions of other applications, or a real-life contact. Furthermore, the fact that they had to log in again to answer these messages was another reason not to use the chat function. GPs mentioned that they did not always communicate with the patient about the results of the eCoach-Pain themselves, but, instead, they asked the PT or PNMH to respond to the patient.


*GP1: “Because of our work-flow, it is the easiest way that the PT communicates with the patient and has a prominent role in the follow-up.”*


They checked if a patient scored a red flag, and only then did they contact the patient or the PT. PT1 discussed the results during nearly each treatment session, while PT2 and the PNMH used a less frequent basis or when the patient had questions about it.

#### 3.3.5. Technical Issues

Six patients did not report any technical issues using the eCoach-Pain. Others mentioned problems in finding how to use all functions of the eCoach-Pain, bugs in sessions, or difficulties connecting their eCoach-Pain to the PHCP’s profile. Two patients perceived difficulties with logging in into the eCoach-Pain because they had to renew their password more than once or had to log in several times in a row. Two patients had help from family or friends with logging in, use of a computer, or receiving reminders. There were no problems mentioned with the instruction manual, and nobody contacted the Helpdesk of the software developer during the pilot period. Four patients registered themselves with the self-subscription option via a website without any problems. The others were registered by their PHCP; in one case, the connection between the application of the patient and the application of the PHCP failed.

Overall, PHCPs indicated that the eCoach-Pain is easy to use. However, all PHCPs reported having difficulties with the two-way factor identification for logging-in, which is obligated by the General Data Protection Regulation (GDPR). They perceived a delay in receiving the codes by email or the email is marked as spam. The fact that there is an extra step for logging-in hindered them in using the eCoach-Pain more often. They also indicated that it is difficult for them to combine the eCoach-Pain with other existing applications in daily practice, as each application has its own login system, function, and layout.

#### 3.3.6. Future Usage and Recommendations of the eCoach-Pain

Most patients were satisfied with the eCoach-Pain. Some patients indicated that the eCoach-Pain supports to increase insight in how pain impacts daily activities and participation and that it answers questions about their pain complaints. Moreover, they recommend it for the use of the chat function with their PHCP. Some patients would recommend the eCoach-Pain because they were satisfied with it themselves. Most of them would recommend it to patients with other complaints than their own, as they indicated that the content did not fit perfectly based on their own situation. They would especially recommend it to patients who are recently diagnosed, have problems in daily activities and participation, are low literate, or who want to use an eCoach-Pain frequently.


*R16: “I would recommend it to people who get acquainted with pain complaints, or who have not so much knowledge yet, for them it is useful to get to know more about pain. But for people who have complaints for years, like me, I would not recommend it.”*


They would not recommend it to patients with pain complaints for years, elderly who are not familiar with eHealth, or patients who do not want to use the eCoach-Pain frequently. However, some patients who are recently diagnosed would recommend it for patients with chronic complaints.

Most PHCPs indicated that they will keep on using the eCoach-Pain in the future as they find it important to offer the patient something extra besides usual care. However, due to time constraints in daily practice, GPs hope that PNMHs can get a more prominent role in the follow-up of patients and contact other PHCPs about the results in the eCoach-Pain. In this case, the PNMH has to contact the GP when expertise or referral of the patient is needed.


*GP1: “Because of the high work-load in primary care, it would be of added value when someone as a PNMH can get a more prominent role in follow-up of patients. They will also be able to keep track of the eCoach-Pain activities. We as GPs have not enough time to do this properly.”*


PTs think they will keep on using the eCoach-Pain in the same way as they did during the feasibility study. However, all PHCPs indicated that the costs of the eCoach-Pain concerning their patient volume are important indicators for future usage. During this feasibility study, these costs were covered by the project budget of NPRL.

## 4. Discussion

The current study provides insight into the feasibility of an eCoach-Pain for patients with CMP or a high risk of becoming chronic, and for PHCPs in interdisciplinary primary care. In general, patients and PHCPs had positive experiences using the eCoach Pain. The answers to questions/statements about learnability, usability, desirability, and adherence to the application confirm that the eCoach Pain has sufficient quality for further use. However, some further adjustments for successful implementation and use are needed.

Some patients mentioned that the content of the eCoach-Pain does not fit with their situation, such as multi-morbidities and previous experiences with treatment. An explanation why patients do not find the eCoach-Pain suitable could be that patients with CMP often experience multi-morbidities, such as depression, anxiety disorders, obesity, hypertension, and diabetes [39,40,41,42]. It has been shown that these patients with multi-morbidities need a personalized treatment [43,44]. The eCoach-Pain has not enough attention for these multi-morbidities. Some patients mentioned that the eCoach-Pain was more suitable for patients with other complaints than they had. Remarkably, the patients with severe complaints for several years mentioned that the eCoach-Pain was better suitable for patients in a subacute phase or those recently diagnosed. Patients with complaints for several years indicated that the information about CMP in the eCoach-Pain was not new and perceived the education sessions as too basic for them. They indicated that the pain education was given in earlier treatments. However, patients with subacute complaints mentioned that the eCoach-Pain is better suited to patients with a clear diagnosis or, in contrast to the comment of patient with long-term complaints, patients with a longer duration of complaints. A possible explanation for this finding could be that subacute patients are still searching for an explanation and solution for their complaints and are, therefore, more biomedically oriented and not yet focused on a biopsychosocial treatment. [45]. As accepting of CMP is an ongoing process, it could be that the patients with subacute musculoskeletal pain do not see themselves as patients with CMP [46]. Therefore, further research is needed to discover for which patients group(s) the eCoach-Pain can be used in primary care and, accordingly, how the eCoach-Pain can be aligned for personalized treatment.

The eCoach-Pain is well integrated into the treatment of the PTs. All PHCPs perceived advantages of the use of the eCoach-Pain during physiotherapy treatment. Patients indicated that eCoach-Pain connects to the treatment of the PT. Positive thoughts about blended rehabilitation care for other diseases are also seen in several other studies [47,48]. The integration of an eCoach as blended physiotherapy care for patients with temporomandibular disorders lead to an increase in self-efficacy, support of data collection and personalization of the application in the Netherlands [47]. The review of Orlando et al. (2019) showed an overall positive impact on patient and caregivers’ satisfaction and it appears to enhance communication and engagement between healthcare professionals for different kinds of telehealth in rural settings [48]. However, questions about the integration of an eCoach in the treatment, such as duration of the treatment, fit in each consultation, and the frequency of the consultations remain [49]. Tilburg et al. recommend to integrate an eCoach into the total treatment and not to implement it as a separate component to the treatment. Further research needs to design and evaluate the integration of the eCoach-Pain into the treatment to deliver blended care.

eCoaches can stimulate and influence interdisciplinary collaboration in primary care [50]. Based on the findings that PHCPs indicated suboptimal collaboration during treatment, it can be concluded that interdisciplinary collaboration between the PHCPs was a point of attention. Accordingly, it seems that the eCoach-Pain did not contribute to interdisciplinary care. GPs indicated that they preferred to refer the patient automatically to a PT as they had not enough time to contact patients and discuss the treatment plan with the PTs, as purposed in interdisciplinary care. The preference of GPs, due to their lack of time, for treatment of these patients by a PT is in line with another study with eCoaches in primary care in the Netherlands [51]. In this study, the PHCPs perceived advantages of the eCoach-Pain in referring a patient to a PT or adding an extra role of a case-manager (for instance, a PNMH or specialized nurses for mental health) in the future. Previous successful implemented eCoaches used a case-manager as first contact for patients [32,34]. In addition, the Standard of Care for Chronic Pain in the Netherlands advices the use of a case-manager for patients with CMP in primary care [14]. However, the eCoaches in these earlier studies were all implemented in secondary care. Therefore, the role for case-manager in primary care needs to be optimized before implementation. Currently, there is no regular financing of a case-manager for patients with CMP in primary care in the Netherlands yet. However, it is crucial to have a case-manager when focusing on integrated and interdisciplinary primary care to stimulate a common vision and treatment plan [19].

Some patients mentioned technical problems that limited the use of diaries or education sessions, even though they received a reminder. Although the developers could not find an explanation for this, it could have influenced the adherence rates for the diaries and education sessions. Other patients indicated that they received too many reminders for diaries. Therefore, in future use of the eCoach-Pain, attention must be given to the communication between the PHCPs and patients. The PHCPs must discuss in advance the number of diaries and reminders offered, based on the preferences of the patients. Research has shown that shared-decision making for chronic illness with treatments containing more than one session leads to treatment agreement [52]. Therefore, shared-decision making in eCoach-Pain adjustments could lead to increased treatment adherence. Connection to the electronic patient file is another technical problem mentioned. PHCPs, and especially GPs, experienced barriers in the use of the eCoach-Pain in daily practice which was not connected with their electronic patient file. This caused double registration steps, which was a reason to restrict use of the eCoach-Pain. Therefore, it would be favored to find a possibility to integrate the eCoach-Pain in the electronic patient file (Dutch: Huisarts Informatie Systeem) to avoid extra registration steps [53].

A major strength of this study is the use of qualitative and quantitative data (mixed-methods) alongside objective data on use of the eCoach-Pain from both patients’ and PHCPs’ perspectives. These data gave a broad overview of the usability of the eCoach-Pain, as well as the experiences. Moreover, the content of the eCoach-Pain was developed together with the PHCPs before the start of the study with a user-centered design [28]. Higgins et al. (2018) recommend user-centered designs and implementation science methods to improve the availability of eHealth tools. [54]. However, the GPs and PNMH rarely used the eCoach-Pain despite their influence in the development process. Reasons for this are the login-facility and lack of time during and after consultations, which are also seen as barriers in the study of Daniëls et al. (2019) in primary care [51].

Some limitations of this study need to be acknowledged. First, the small sample of patients that were available for this study and the limited use of the eCoach-Pain could have introduced selection bias. It could be that patients with, for example, low literacy or co-morbidities, were not asked for participation by the PHCPs with a risk for selection bias. However, despite the small sample, patients differed in demographic characteristics, resulting in sufficient confidence to have studied a representative group of users. Second, not all patients performed all measurements, so the completeness of available data per measurement differed. Six patients did not respond on the evaluation questionnaires, and, for three patients export data of eCoach-Pain-use is missing. This could have led to information bias which could have influenced the data. Third, the sample of primary care practices and PHCPs was small, which could have influenced the results. As for primary care practice 1, all patients are subscribed by a GP or PNMH, and, for primary care practice 2, the PT also subscribed patients, besides the GP and PNMH. Most patients were recruited by primary care practice 2 (*n* = 21), and most of the patients participating in the interviews were also recruited by this practice (9 out of 11). Therefore, limited results were available about the recruitment of GPs in the interviews.

## 5. Conclusions

In conclusion, the eCoach-Pain seems to be promising in primary care: the patients, as well as the PHCPs, experienced advantages for treatment of patients with CMP. However, adjustments to the content have to be made for better fit with patient-specific CMP complaints. Moreover, the implementation strategy seems to be an important factor for successful use among PHCPs. This should be improved for successful use in interdisciplinary primary care settings. The involvement of a case-manager for CMP should be further explored when implementing the eCoach-Pain. Thereby, it is important to use user-centered designs and implementation science methods to evaluate adjustments resulting in a successful implementation.

## Figures and Tables

**Figure 1 ijerph-18-11661-f001:**
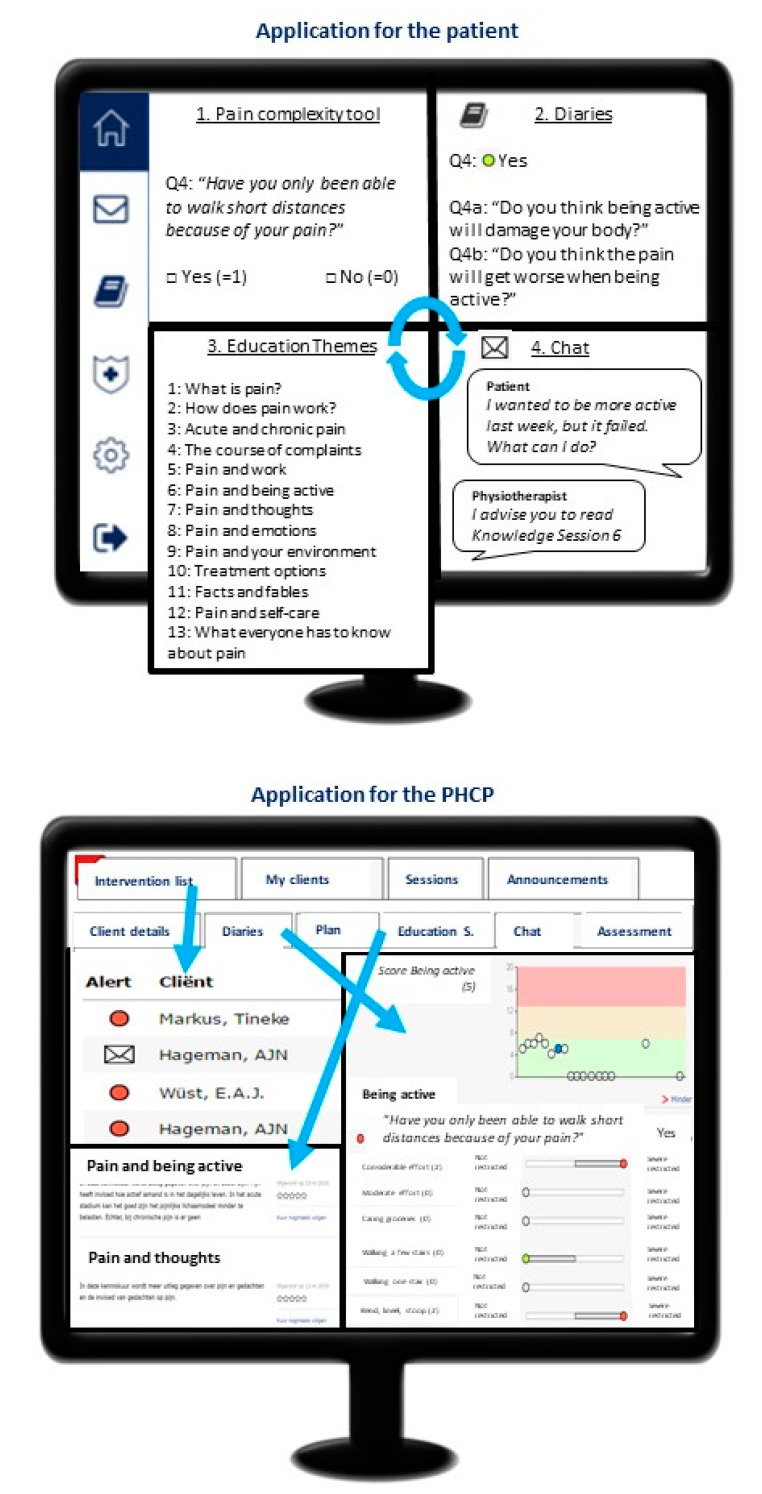
Content of the eCoach-Pain. At the top is the application for the patient, with the pain complexity tool, diaries, educational sessions, and chat function. At the bottom is the application for the primary healthcare professional (PHCP) with an intervention list, an overview of the diaries, and an overview of educational sessions.

**Table 1 ijerph-18-11661-t001:** Patient characteristics.

					Registration via		Primary Care Practice		Data Available		
Participant Number	Sex (%)	Age (y) (Median, IQR)	Days Active in the Coach Till Export (Median, IQR)	First Contact Person	PHCP (*n*, %)	Webpage (*n*, %)	1 (*n*, %)	2 (*n*, %)	Questionnaire (*n*, %) September 2020	Export (*n*, %) October 2020	Interview (*n*, %) September + December 2020
R01	M	41	103	GP	X		X		X	X	
R02	M	61	86	GP		X	X		X	X	X
R03	M	36	124	GP		X	X		X	X	
R04	F	69	109	PT		X		X	X	X	
R05	M	70	54	GP		X		X	X	X	
R06	M	71	115	GP		X		X	X	X	X
R07	F	67	133	PT	X			X	X	X	X
R08	M	70	61	GP		X	X			X	
R09	F	43	150	PT	X			X	X	X	
R10	F	65	7	PT		X		X	X	X	X
R11	F	66	112	GP		X		X	X	X	
R12	F	47	68	PT	X			X	X	X	
R13	F	46	117	PT	X			X		X	X
R14	F	60	92	GP		X	X			X	X
R15	M	62	117	GP		X		X		X	X
R16	F	32	83	GP		X		X		X	X
R17	F	50	n.a.	PNMH	X			X			X
R18	F	41	n.a.	PT	X			X			X
R19	F	45	n.a.	PT	X			X			X
R20	M	50	65	PT	X			X		X	
R21	F	57	98	GP		X	X			X	
R22	F	35	56	PT	X			X		X	
R23	F	44	105	GP		X		X		X	
R24	F	24	127	GP		X	X			X	
R25	F	29	118	PNMH	X		X			X	
R26	F	64	115	GP		X		X		X	
R27	F	40	59	PT	X			X		X	
R28	F	38	144	PT	X			X		X	
R29	F	65	127	PT		X		X		X	
Total:	F:21 72%	50.0—24.0	107.0—46.0	GP: 14, PT: 13, PNMH: 2	13 45%	16 55%	8 28%	21 72%	11 38%	26 90%	11 38%

F: female, M: male, GP: general practitioner, PT: physiotherapist, PNMH: practice nurse mental health. n.a.: not applicable. IQR: interquartile range, *n*: total number, %: percentage of total.

**Table 2 ijerph-18-11661-t002:** Median (IQR) learnability, usability, and desirability scores for the total patient group, patients subscribed by GPs, and patients subscribed by PTs.* (1–5).

	Subscribed by		
	GP and PT (*n* = 11)	GP (*n* = 6)	PT (*n* = 5)
Learnability	5.0 (1.0)	5.0 (0.0)	5.0 (1.0)
It was easy to learn how to use the application.	5.0 (0.5)	5.0 (0.0)	5.0 (1.0)
I think the application was very complicated. ^a^	5.0 (0.5)	5.0 (0.5)	5.0 (1.0)
I needed a lot of help to learn using the application. ^a^	5.0 (0.5)	5.0 (0.0)	5.0 (3.0)
I quickly caught on how I could use the application.	5.0 (1.0)	5.0 (0.8)	4.0 (1.0)
I am confident that I used the application in the right way.	5.0 (1.5)	5.0 (0.8)	4.0 (0.8)
Usability	5.0 (1.5)	5.0 (1.0)	5.0 (2.0)
I could easily login into the application.	5.0 (1.0)	5.0 (0.8)	4.0 (2.0)
I could easily report my pain, activities, feelings, thoughts, and emotions.	5.0 (1.5)	5.0 (0.8)	5.0 (3.0)
I understood the information in the educational sessions about my pain, activities, feelings, thoughts, and emotions.	5.0 (1.0)	5.0 (0.8)	5.0 (2.0)
I could easily search for information about pain with the application.	5.0 (2.0)	5.0 (1.5)	4.0 (2.0)
I could easily leave a message for the PHCP via the application.	5.0 (1.5)	4.5 (1.8)	5.0 (0.0)
Desirability	4.0 (2.0) (*n* = 10)	4.5 (2.0)	4.0 (1.0) (*n* = 4)
I liked using the application.	4.0 (1.0)	5.0 (1.5)	4.0 (1.8)
I liked using the pain diary for reporting my pain, activities, feelings, thoughts, and emotions.	4.0 (1.0)	5.0 (0.8)	4.0 (1.3)
I liked using the educational sessions.	4.0 (1.8)	4.5 (2.0)	3.5 (1.0)
I liked using the chat function.	3.0 (1.8)	3.5 (1.8)	3.0 (1.0)
I liked the idea that my PHCP monitors my pain, activities, feelings, thoughts, and emotions.	4.5 (1.0)	4.5 (1.0)	4.5 (1.5)
I liked the idea that my PHCP could adjust my treatment based on my answers in the eCoach-Pain.	4.0 (2.0)	4.0 (1.8)	4.0 (2.3)
I would like to recommend the application to other patients.	4.0 (2.0)	4.5 (1.8)	3.5 (1.0)
Total overall score (0–10)	7.0 (2.8)	8.5 (2.3)	6.5 (0.5)

Scores: 0—totally disagree, 5 or 10—totally agree. ^a^ Negatively-keyed items were reversed-scored before data analyses but the original question is presented in the table with the reversed-score. * PNMHs have no patients subscribed which completed the questionnaire.

## Data Availability

The data presented in this study are available on request from the corresponding author. The data are not publicly available due to the small number of healthcare professionals and patients included. This could interfere with the privacy of the participants.

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
