# Peer review of "An eCoach-Pain for Patients with Chronic Musculoskeletal Pain in Interdisciplinary Primary Care: A Feasibility Study"

_ijerph, 2021, doi:10.3390/ijerph182111661_

Round 1

Reviewer 1 Report

Thank you for the opportunity to review the manuscript “An eCoach-Pain for patients with chronic musculoskeletal pain in interdisciplinary primary care: a feasibility study”. My comments are below. I hope they will be useful to the authors.

Specific comments:

  • Abstract: Please insert how many primary health care professional (PHCP) that participated in the workshop and how long (time) participants had access to eCoach-Pain.
  • Introduction (line 47). The references 11 and 13 are from pediatric settings. Do you have other references for adults?
  • Introduction (line 67). Please insert more updated references. There are more recent reviews that can support this statement.
  • Methods: Do patients have to pay to get access to follow up by the Network Pain Rehabilitation Limburg? Did the eCoach–Pain have costs?
  • Methods: For how long did the patients have access to eCoach–Pain before extraction of use data?
  • 2.2.2. How did PHCPs access the eCoach–Pain. Did they have access at the same computer system they were working at or did they have to log on to a new place?
  • Methods (line 148). Consider moving “B)” to a new line like “A” at the previous page.
  • Methods (line 182): This heading is different than the 2.2.1-heading
  • Methods (line 219). What disciplines are you referring to? PHCP?
  • Methods (line 226-230). Usage data are often not normally distributed, and therefore median and the corresponding range are better to describe data. Was your data normally distributed?
  • Methods (line 242). Please add a reference to the thematic analysis, and describe some about how you conducted the thematic analysis. Inductive? Deductive? How did you categorize/sort the data?
  • Results: In the text (line 263) you use the term “men” and “women”. In table 1 you use the term “male” and “female”. Please be consistent in the whole manuscript.
  • Results, Table 1: Please consider revising the table a bit. It is hard to follow that the line in dark grey are a summary of the above. Do you need all the details? Or could you just have a table with the summary scores?
  • Results (line 275-282). Please avoid repetition of results (e.g., responder rate: 57.8, proportion of female responders).
  • Results, Table 2. You have few respondents, and I wonder if the data were normally distributed? Could median and range be a better way to show the results?
  • Results, Table 2. What is the “total” in the last line of the table? You describe something in line 296, but it is difficult to understand what “total” measures
  • Results (line 306): How was the average 97.7 days active calculated? Days from first to last use? Individual days used?
  • Results (line 314-315). You state: “Each educational session was opened by on average 18.7 (SD=3.1) patients”. Please consider rephrasing. Is it 18.7 times?
  • Results (line 318). I would recommend to insert a percentage behind ”seven” to make it easier for the reader to see the proportion of users.
  • Results (line 325-227): This does not correspond to what you are writing in line 234-237 in the methods.
  • Results (line 481): Should “FT” be replaced with “PT”?
  • References:
    • 7: Please insert an URL or a more specific reference
    • 14: Who is the publisher/author? Please update the date of access

Reviewer 2 Report

Overall a great evaluation of an interesting product that I'm sure will help many patients, it has a lot of potential uses.  Additionally, I must commend you for your in-depth description of the process of using the application!

The only comments that I have relate to wording / grammar concerns, please see below 

Introduction

  1. Line 36-38: "calls for a change in health systems focusing on interdisciplinary rehabilitation care over the whole health system, focusing on the im provement of self-management skills of patients on long term" --> I understand what you are trying to say, but this seems a bit wordy
  2. Line 52 - You mention primary, secondary, and tertiary care; could you perhaps expand a bit on what the distinction between these are to readers outside of your Country.
  3. Line 78 - You mention earlier feedback - is this from the previously mentioned studies? Otherwise its not clear where this earlier feedback is coming from. 

Materials and Methods  

  1. Line 112 - "... risk of becoming chronic consulted" --> Perhaps there is a word missing or a typo between chronic and consulted

Results

  1. Line 262 - "Data were..." --> change were to was
  2. Lines 262 - 267 - you are not consistent with wording of numbers, in one instance you have the number 2 and in another instance you type out the word eight
  3. Line 301 - "the export data (only exported once)" --> what are you trying to say here? Are you just referring to the exported data, in which case you can simply state the exported data , otherwise what is the significance of it being exported more than once
  4. Line 322 - "35,5 times" --> elsewhere you use a period to denote a decimal place
  5. Line 352 - "some found it frustrating..." --> can you quantify how many that is?
  6. Line 363 - "but did less use the results" --> not sure what you mean here
  7. Line 436 - 437 - ...and acceptance in the sessions...one patient indicated that the sessions were confronting." --> not sure what you mean by what are patients accepting and confronting?
  8. Lines 516 - 521 - you start off the sentence by stating "however some patients who were recently diagnosed would recommendit for patients with chronic complaints", however the following quote is from a chronic patient recommending it for new patients, so it is a bit confusing

Discussion

  1. Line 539 - "application were on average rated as good" - what does good mean? Your scale was from totally disagree to agree - so what quantifies as good?
  2. Line 631 - 632 - you discuss selection bias - can you expand a little bit on this because I am not sure what selection bias you are talking about 
